# Control of Grapevine Downy Mildew by an Italian Copper Chabasite-Rich Zeolitite

**Francesco Calzarano** [1,*] **, Leonardo Seghetti** [2] **, Giancarlo Pagnani** [1] **, Elisa Giorgia Metruccio** [3] **and Stefano Di Marco** [3]

1. Faculty of Bioscience and Technologies for Food, Agriculture and Environment, University of Teramo, 64100 Teramo, Italy; gpagnani@unite.it
2. IIS "C. Ulpiani", 63100 Ascoli Piceno, Italy; leonardo.seghetti@tin.it
3. Italian National Research Council (CNR), Institute of BioEconomy (IBE), 40129 Bologna, Italy; elisa.metruccio@ibe.cnr.it (E.G.M.); stefano.dimarco@ibe.cnr.it (S.D.M.)
* Correspondence: fcalzarano@unite.it

**Abstract:** The progressive reduction in the quantities of copper regulated by the European Union is focusing the research on new formulations with a reduced copper content but equally effective. In this regard, the activity of an Italian copper chabasite-rich zeolitite, which proved to be effective against grapevine grey mold and sour rot, was assessed against downy mildew. A two-year study was carried out in the Abruzzo region, Italy, in a cv. Montepulciano vineyard. The applications of the copper zeolitite showed the same good results obtained by a standard integrated/conventional strategy based on contact and systemic fungicides. At harvest, in both trial years, the plants with infected bunches in the untreated control ranged from 86.25% to 100%, compared to 15–30% of the treated plants. Furthermore, infected bunches and berries of the untreated control vines were 70–100% while treated ones never exceeded 2.32%. Furthermore, an increase in the polyphenol content and color intensity in wines made from vines treated with copper zeolitite was confirmed and appeared to be particularly evident in hot and dry seasons. The activity of copper zeolitite towards downy mildew, the potential use against grey mold and sour rot and the protection of grapes from high temperatures indicate that this product is a promising tool for a viticulture environmentally friendly control strategy.

**Keywords:** *Plasmopara viticola*; *Vitis vinifera* L.; low environmental impact fungicides; natural substances; sustainable control strategy

## 1. Introduction

*Plasmopara viticola* (Berk. and Curt.) Berl. and de Toni, the causal agent of grapevine downy mildew, is a highly destructive pathogen all over the grape-growing areas [1]. Buds, leaves, flowers and berries are attacked by the disease, particularly under high rainfall and humidity conditions, with prolonged wetness on leaf and berry [2,3].

The adaxial leaf surface is affected by chlorotic oil spots. The sporulation of sporangiophores and sporangia occurs in the abaxial surface of the leaf, causing a number of secondary infections in relation to the seasonal rains [4,5].

The disease causes the most serious yield losses affecting inflorescences, bunches and berries [6]. Early downy mildew infections can occur from pre-flowering to the end of flowering, by penetrating the stomata with germinative tubules of zoospores. In the presence of high humidity conditions, the grape cluster is completely covered by a whitish mildew caused by fungal sporulation.

Until the end of flowering, the infection can lead to a deformation of clusters, which can become dry. From flowering to fruit set, as long as the berries do not exceed 2.5 mm in diameter, berries can be still affected by downy mildew through stomatal penetration. Later, when stomata become atrophied, the pathogen can only enter the berries through the

peduncle. Berries dehydrate and turn brown, without showing any reproductive organs of the fungus. This symptomatic expression is known as berry brown rot [5].

The control of the disease is based on different contact and systemic fungicides. Contact fungicides are mainly based on copper and dithiocarbamates and can protect the foliar surface by preventing the zoospore germination and leaf penetration [7,8].

Fungicides able of penetrating inside plant tissues are characterized by a different degree of penetration and movement into the plant. Among translaminar products, cymoxanil showed the best translaminar characteristics [9].

The systemic fungicides, such as phenylamides and fosetyl-Al, can penetrate the plant tissues and are translocated through the xylem or both xylem and phloem, respectively, also protecting the plant far from the site of application [10]. Further systemic fungicides, such as mitochondrial respiration inhibitors (Qoi), e.g., famoxadone and fenamidone, are active in cellular respiration with a site-specific mode of action [10,11].

Chemical fungicides caused environmental contaminations and negative human health effects [12]. Moreover, several groups of single-site fungicides have caused resistance to downy mildew pathogen strains, thus losing effectiveness against the pathogen they must control [13]. All these reasons led to the progressive withdrawal of chemical fungicides and the need to carry out studies on new safe and low-impact fungicides [14].

The development of products based on microorganisms in grapevine disease control increased, but in many cases, these products have provided inconsistent and not always effective results against *P. viticola*. Moreover, they can be used in the replacement of one or a few more treatments with chemical formulations.

The development and introduction of inert natural substances in the vine control strategies, such as rock flours, could overcome the limits described above in the use of antagonistic microorganisms. Possibly, inert natural substances should be characterized by a broad spectrum of activity against different diseases. In fact, the introduction of a new product in disease control is easier and more functional if its activity can be explained on different hosts and pathogens [15].

Recently, in vineyards of the Abruzzo region, central Italy, chabasite-rich zeolitites, both pure and with copper additions, were effectively applied for the concomitant control of grey mold, sour rot and grapevine moth [16,17].

Zeolitites are rocks characterized by a high content of pure zeolite (>50%). Zeolites belong to a mineral family consisting of 52 different species structurally belonging to tectosilicates such as silica minerals, feldspar and feldspathoids [18]. The main characteristics of these minerals that make them useful in different applications in agriculture are cation exchange capacity (CEC), hydration properties and adsorption capacity [19]. Thanks to these properties, zeolitites improve soil physicochemical characteristics, such as (i) increasing CEC and water retention, maintaining permeability in sandy soils and increasing aeration and permeability in clayey soils; (ii) temporary capture by cationic exchange of K and N-ammonia supplied by a synthetic or natural fertilizer, and their slow and gradual release; and (iii) greater availability of nutrients for crops and lower losses due to leaching [20]. In particular, some zeolitites, due to the high exchange capacity of the ammonium ion, are used to prevent the microbial oxidation of NH4-N in the leachable NO3-N form [21,22].

Chabasite-rich zeolitites, widely spread in central Italy (Lazio and Tuscany regions), have high values of CEC (2.1–2.2 meq/g) and water retention (40–45% *w/w*). Therefore, these zeolitite properties have pushed the evaluation of the activity of a chabasite-rich zeolitite in the simultaneous control of the grapevine diseases and pests described above [16,17].

On the basis of previous encouraging results against grey mold and sour rot, the present study, carried out in a vineyard of Abruzzo region, investigated the activity of a copper chabasite-rich zeolitite for the control of *P. viticola*. The use of a formulation with a reduced copper content compared to other traditional cupric formulations is an added value of the product, especially because of the restriction of the use of copper compounds in the vineyard [23,24].

To our knowledge, this is the first study concerning the use of a copper chabasite-rich zeolitite for the control of grapevine downy mildew. For each of the two years of the investigation, the study compared a standard integrated/conventional strategy, normally carried out in the vineyard, with a strategy based only on copper chabasite-rich zeolitite applications. Furthermore, evaluations on the quality of yield and wines of the different treatments were carried out.

## 2. Materials and Methods

### 2.1. Vineyard Trial

Trials were carried out in 2015 and 2016 in a Montepulciano (clone VCR 100) 16-year-old vineyard, rootstock 1103 P, bilateral Guyot trained with $2.5 \times 1$ m planting and located in Ari, Chieti, Central Italy. The vineyard, of about 5000 m$^2$, is located on the plain area under climatic conditions favorable to downy mildew infections. Three treatments were compared: (1) copper chabasite-rich zeolitite, (2) fungicides normally used in the vineyard towards downy mildew and (3) untreated control. Each treatment included 3 replicates, each consisting of plots of 500 m$^2$. The field experiment was set up as randomized block design (Tables 1 and 2).

Following the weather forecast, chabasite-rich zeolitite applications were carried out before each rainfall associated with a possible infection, taking into account the characteristics of the compound as adhesiveness, persistence and rainfastness. Similarly, a preventive strategy was adopted for the fungicides normally used on the farm, including systemic fungicides. An agrometeorological station (DigitEco s.r.l., Bologna, Italy) in the vineyard collected the climatic parameters (Tables 1 and 2; Tables S1 and S2).

The zeolitite used in this study comes from quarries in Sorano (Grosseto), central Italy (Agrisana s.r.l., Latina, Italy). After extraction, by means of hydraulic or pneumatic drilling machines, the product is subjected to an industrial process to incorporate into the mineral structure the copper, such as sulfate pentahydrate and hydroxide.

The copper metal content of the product is 6%. Therefore, for each application with copper zeolitite at 4 kg ha$^{-1}$, the amount of copper metal was 240 g ha$^{-1}$.

For each application of Bordeaux mixture + copper sulphate complexed with amino acids, the amount of copper was 562 g ha$^{-1}$ (20% of copper metal in Bordeaux mixture applied at 2.5 kg ha$^{-1}$ and 6.2% of copper metal in copper sulphate complexed with amino acids applied at 1 kg ha$^{-1}$).

The mineralogical composition of the copper chabasite-rich zeolitite used in the present study has been reported in Calzarano et al. (2019) [16].

Field trial details such as dates of compound applications, growth stages, formulations, doses of use and rainfall occurred after the compound applications are shown in Tables 1 and 2.

The compound applications were carried out with a pneumatic sprayer supplied by the farm and calibrated for a volume of 500 L ha$^{-1}$.

**Table 1.** Field application details targeted against grapevine downy mildew in cv. Montepulciano in the Ari vineyard in 2015 and rainfall amount on the days after the applications.

| Application Number | Application Date | BBCH Growth Stages | Treatment | Dose (Kg/L ha$^{-1}$) | Rainfall | |
|---|---|---|---|---|---|---|
| | | | | | Days | Total Amount (mm$^{-1}$) |
| 1 | 18 May | 57 | (1) Italian copper chabasite-rich zeolitite<br>(2) Fenamidone + Fosetyl Al + Iprovalicarb | 4<br>2.5 | 19–28 May | 132.1 |
| 2 | 11 June | 71 | (1) Italian copper chabasite-rich zeolitite<br>(2) Cymoxanil + Mancozeb | 4<br>3 | 12–15 June | 11.1 |
| 3 | 16 June | 73 | (1) Italian copper chabasite-rich zeolitite<br>(2) Fenamidone + Fosetyl Al + Iprovalicarb | 4<br>2.5 | 17–20 June<br>24 June | 38.8<br>30.5 |
| 4 | 6 August | 81 | (1) Italian copper chabasite-rich zeolitite<br>(2) Bordeaux mixture + copper sulphate complexed with amino acids | 4<br>2.5 + 1 | 7 August | 75.1 |
| 5 | 13 August | 83 | (1) Italian copper chabasite-rich zeolitite<br>(2) Bordeaux mixture + copper sulphate complexed with amino acids | 4<br>2.5 + 1 | 15–16 August | 41.4 |
| 6 | 3 September | 85 | (1) Italian copper chabasite-rich zeolitite<br>(2) Bordeaux mixture + copper sulphate complexed with amino acids | 4<br>2.5 + 1 | 4–8 September | 13.7 |
| 7 | 19 September | 85 | (1) Italian copper chabasite-rich zeolitite<br>(2) Bordeaux mixture + copper sulphate complexed with amino acids | 4<br>2.5 + 1 | 20–26 September | 107.0 |

BBCH: Biologische Bundesanstalt, Bundessortenamt und Chemische Industrie. Active ingredient concentration (%): Fenamidone + Fosetyl Al + Iprovalicarb: 4 + 52 + 4.8; Cymoxanil + Mancozeb: 4 + 40; Bordeaux mixture: 20% copper; Copper sulphate complexed with amino acids: 6.2% copper.

**Table 2.** Field application details targeted against grapevine downy mildew in cv. Montepulciano in the Ari vineyard in 2016 and rainfall amounts on the days after the applications.

| Application Number | Application Date | BBCH Growth Stages | Treatment | Dose (Kg/L ha$^{-1}$) | Rainfall | |
|---|---|---|---|---|---|---|
| | | | | | Days | Total Amount (mm$^{-1}$) |
| 1 | 30 April | 19 | (1) Italian copper chabasite-rich zeolitite<br>(2) Cymoxanil + Mancozeb | 4<br>3 | 01–04 May | 45.3 |
| 2 | 9 May | 53 | (1) Italian copper chabasite-rich zeolitite<br>(2) Cymoxanil + Mancozeb | 4<br>3 | 11–12 May<br>15–17 May | 10.4<br>28.9 |
| 3 | 18 May | 57 | (1) Italian copper chabasite-rich zeolitite<br>(2) Cymoxanil + Mancozeb | 4<br>3 | 19–20 May | 29.6 |
| 4 | 31 May | 60 | (1) Italian copper chabasite-rich zeolitite<br>(2) Fenamidone + Fosetyl Al + Iprovalicarb | 8<br>2.5 | 1–5 June | 30.2 |
| 5 | 8 June | 71 | (1) Italian copper chabasite-rich zeolitite<br>(2) Fenamidone + Fosetyl Al + Iprovalicarb | 4<br>2.5 | 9–10 June<br>12–14 June | 17.6<br>31.2 |
| 6 | 17 June | 73 | (1) Italian copper chabasite-rich zeolitite<br>(2) Bordeaux mixture + copper sulphate complexed with amino acids | 4<br>2.5 + 1 | 19–20 June | 15.8 |
| 7 | 13 July | 77 | (1) Italian copper chabasite-rich zeolitite<br>(2) Bordeaux mixture + copper sulphate complexed with amino acids | 4<br>2.5 + 1 | 15–16 July | 148.4 |
| 8 | 30 July | 79 | (1) Italian copper chabasite-rich zeolitite<br>(2) Bordeaux mixture + copper sulphate complexed with amino acids | 4<br>2.5 + 1 | 1 August | 14.2 |
| 9 | 22 August | 83 | (1) Italian copper chabasite-rich zeolitite<br>(2) Bordeaux mixture + copper sulphate complexed with amino acids | 4<br>2.5 + 1 | 23 August<br>31 August | 0.4<br>1.2 |
| 10 | 4 September | 85 | (1) Italian copper chabasite-rich zeolitite<br>(2) Bordeaux mixture + copper sulphate complexed with amino acids | 4<br>2.5 + 1 | 5–12 September | 44.2 |
| 11 | 15 September | 85 | (1) Italian copper chabasite-rich zeolitite<br>(2) Bordeaux mixture + copper sulphate complexed with amino acids | 4<br>2.5 + 1 | 16–21 September | 18.2 |

BBCH: Biologische Bundesanstalt, Bundessortenamt und Chemische Industrie. Active ingredient concentration (%): Fenamidone + Fosetyl Al + Iprovalicarb: 4 + 52 + 4.8; Cymoxanil + Mancozeb: 4 + 40; Bordeaux mixture: 20% copper; Copper sulphate complexed with amino acids: 6.2% copper.

## 2.2. Disease Assessment

In both trial years, 4 assessments to evaluate the downy mildew infections in all treatments were carried out on 6 June (BBCH, Biologische Bundesanstalt, Bundessortenamt und Chemische Industrie, 60), 2 July (BBCH 75), 14 August (BBCH 83) and 3 October 2015 (BBCH 89); 30 May (BBCH 60), 30 June (BBCH 75), 10 August (BBCH 83) and 30 September 2016 (BBCH 89).

The first two assessments of the season were aimed at evaluating the downy mildew symptoms on the leaves and possible damage to bunches. The next two assessments evaluated the infection on bunches. In each of the three plots of each treatment, 80 plants from the central area of the plot were selected. The percentage of plants with infected leaves (first and second assessments) or bunches (third and fourth assessment) were recorded.

In the first two assessments, in each of the 80 plants of each plot, 50 leaves were randomly selected and observed for the presence of downy mildew symptoms. The percentage of infected leaves per plant, infected surface per leaf and surface with zoosporangia per leaf were calculated. In the third and fourth assessment, in the same plants of each treatment, the percentage of bunches affected by downy mildew was calculated by comparing the number of infected bunches with the total bunches of the plant. Each plant had a uniform number of 16 bunches.

Furthermore, in each plant the percentage of infected berries was calculated as follows. For each treatment, the berries of 100 clusters were assessed and the average number of berries per cluster was calculated. The average number of berries per plant was calculated by multiplying the average number of berries by the number of clusters of the plant (16). For each treatment, the percentage of infected berries per plant was obtained by comparing the number of infected berries of the plant to the average number of berries of the plant.

## 2.3. Effect of the Applications against Downy Mildew on Yield Quantity, Grape and Wine Composition

At harvest, on 4 October 2015 and 5 October 2016, the yield was assessed in each of the plots of each treatment by weighing the bunches collected from 6 plants of the central area of each plot. In 2015, the yield of 6 untreated plants not infected by the disease was assessed. On the contrary, in 2016, in the untreated control plots it was not possible to carry out this assessment since all bunches were infected by downy mildew.

On the same days in which the harvest was carried out, from each plot of each treatment 100 berries were collected, thus obtaining 3 replicates of 100 berries (one per plot) for each treatment. As reported above for the assessments of yield in the control plots, in 2016 it was not possible to collect the berries, which in 2015 were collected from plants that remained healthy until harvest. The berries were collected alternately from the wings, tips and center of the bunch. The berry samples were analyzed for soluble solids ($^\circ$ Brix), pH, total acidity and total polyphenols according to the methods: Fheling, potentiometric, acid/base titration (g $L^{-1}$) and Folin–Ciocalteau (mg $L^{-1}$), respectively.

Moreover, in both trial years, for each plot, from each of the 6 plants (except for the untreated control plants in 2016), a sample of berries (0.5 $Kg^{-1}$) was collected. Therefore, 3 replicates of 3.0 $Kg^{-1}$ per treatment were obtained.

Immediately after harvesting, the berries were crushed with a manual press. The samples were vinified according to the methodology described in Calzarano et al. (2019) [16]. At the end of the vinification, about 2.1 $L^{-1}$ of wine was obtained from each 3.0 $Kg^{-1}$ sample. After three months of ageing, each wine sample was analyzed for ethyl alcohol, pH, total acidity, total polyphenols and wine color intensity (IC3) according to the following methods: distillation, potentiometric, acid/base titration (g $L^{-1}$) Folin–Ciocalteau (mg $L^{-1}$) and spectrophotometric, respectively.

Berry and wine analyses were carried out following the methods of the Official Gazette of the European Communities Regulation (EEC) No. 2676/90, Official Journal L 272, 3.10.1990 [25].

## 3. Results

### 3.1. Vineyard Trial

In the Ari vineyard, the growth seasons 2015 and 2016 were characterized by the occurrence of many downy mildew infections as a result of heavy rainfalls. In the two trial years, the amount of rainfall from sprouting to harvest was similar, 466 and 458 mm$^{-1}$ in 2015 and 2016, respectively. However, the higher rainfall frequency in 2016 caused a higher number and an increased severity of infections compared to 2015, thus requiring more fungicide applications (Table 2 and Table S2).

#### 3.1.1. The 2015 Trial

In 2015, the first infections in the untreated control occurred after 132.1 mm$^{-1}$ of rainfall recorded from 19 to 28 May (Table 1 and Table S1). In the first assessment of June 6 (BBCH 60), 7.08% of untreated vines showed symptoms (oil spot) on the leaves (Table 3). However, the percentage of symptomatic leaves was very low (0.18%), as for the percentage of infected surface and fertile infected surface (percentage leaf area with sporulating lesions) of symptomatic leaves (0.15% for both parameters). In the second assessment (July 2, BBCH 75) after three rainy periods in June (Table 1 and Table S1), the percentage of symptomatic plants in the untreated control was 60.83%. The percentages of symptomatic infected leaves (3.48%), of infected surface (4.93%) and of fertile infected surface (4.93%), also increased. In the first two assessments, no damage of the bunches was ever noticed (Table 3).

In the first and second assessments, both chabasite-rich zeolitite and farm products were effective towards downy mildew infections by significantly reducing all the investigated parameters compared to the untreated control ones (Tables 1 and 3). The percentage of treated vines showing leaf symptoms recorded in the second assessment (July 2) was 2.92 and 2.08% for chabasite-rich zeolitite and farm products, respectively. Furthermore, sporulation from the few infected leaves was rarely observed in the treated plants and was significantly lower ($p = 0.05$) compared the untreated control (Table 3).

In the third assessment (August 14, BBCH 83), after 75.1 mm$^{-1}$ rainfall of August 7, 74.17% of the untreated control vines showed symptoms of berry brown rot (Table 4). Moreover, the percentages of infected bunches and infected berries of the untreated control vines were 12.89 and 7.43%, respectively, both significantly higher ($p = 0.05$) compared to the percentages of the treated vines (Table 4). In the last assessment (October 3), at harvest maturity (BBCH 89), after the rainfalls in August and September, the percentage of plants with infected bunches in the untreated control was 86.25% (Table 4). The percentages of infected bunches (75.23%) and infected berries (70.33%) greatly increased, compared to what was observed in the previous assessment (Table 4).

Both chabasite-rich zeolitite and farm product applications carried out from veraison to harvest effectively protected the plant, especially considering the high levels of infection in the untreated control. In the third and fourth assessments, similar to the first two, the values of the parameters detected in chabasite-rich zeolitite did not statistically ($p = 0.05$) differ from the farm products ones. The percentages of vines with infected bunches did not exceed 10% (third assessment) and 15% (fourth assessment) (Table 4). Contrary to what was observed in the untreated control, the percentage of infected bunches and infected berries slightly increased from the third to the fourth assessment, with values, at harvest maturity, of 1.22% and 1.43% and 0.55% and 0.61% in the plots treated with chabasite-rich zeolitites and farm products, respectively (Table 4).

**Table 3.** Comparison of incidence and severity among control strategies of grapevine downy mildew based on applications of Italian copper chabasite-rich zeolitite or fungicides provided for the farm plan in the Montepulciano cultivar of Ari vineyard: leaf surveys.

| Survey | Treatment | * Infected Plants | Infected Leaves | Infected Leaf Surface | ** Infected Leaf Fertile Surface |
|--------|-----------|:---:|:---:|:---:|:---:|
| | | (%) | | | |
| 06/06/2015 | 1—Copper chabasite-rich zeolitite | 0.42 b | 0.01 b | 0.01 b | 0.00 b |
| | 2—Farm | 0.42 b | 0.02 b | 0.01 b | 0.00 b |
| | 3—Untreated control | 7.08 a | 0.18 a | 0.15 a | 0.15 a |
| 02/07/2015 | 1—Copper chabasite-rich zeolitite | 2.92 b | 0.06 b | 0.04 b | 0.01 b |
| | 2—Farm | 2.08 b | 0.05 b | 0.04 b | 0.01 b |
| | 3—Untreated control | 60.83 a | 3.48 a | 4.93 a | 4.93 a |
| 30/05/2016 | 1—Copper chabasite-rich zeolitite | 1.25 b | 0.03 b | 0.02 b | 0.00 b |
| | 2—Farm | 1.25 b | 0.03 b | 0.05 b | 0.01 b |
| | 3—Untreated control | 11.67 a | 0.33 a | 0.51 a | 0.51 a |
| 30/06/2016 | 1—Copper chabasite-rich zeolitite | 4.17 b | 0.08 b | 0.06 b | 0.03 b |
| | 2—Farm | 3.75 b | 0.09 b | 0.10 b | 0.05 b |
| | 3—Untreated control | 89.17 a | 4.88 a | 5.26 a | 5.26 a |

* Percentage of plants with infected leaves. ** Percentage leaf area with sporulating lesions. Statistical analyses were performed according to Tukey's honest significant difference (HSD) test. For each column, assessment and year, different letters represent significant differences at $p = 0.05$.

**Table 4.** Comparison of incidence and severity among control strategies of grapevine downy mildew based on applications of Italian copper chabasite-rich zeolitite or fungicides provided for the farm plan in the Montepulciano cultivar of Ari vineyard: grape surveys.

| Survey | Treatment | * Infected Plants | Infected Bunches | Infected Berries |
|---|---|---|---|---|
| | | (%) | | |
| | 1—Copper chabasite-rich zeolitite | 9.17 b | 0.70 b | 0.25 b |
| 14/08/2015 | 2—Farm | 7.03 b | 0.57 b | 0.17 b |
| | 3—Untreated control | 74.17 a | 12.89 a | 7.43 a |
| | 1—Copper chabasite-rich zeolitite | 12.92 b | 1.22 b | 0.55 b |
| 03/10/2015 | 2—Farm | 14.58 b | 1.43 b | 0.61 b |
| | 3—Untreated control | 86.25 a | 75.23 a | 70.33 a |
| | 1—Copper chabasite-rich zeolitite | 18.33 b | 1.38 b | 0.32 b |
| 10/08/2016 | 2—Farm | 21.67 b | 1.56 b | 0.22 b |
| | 3—Untreated control | 100.00 a | 30.36 a | 9.46 a |
| | 1—Copper chabasite-rich zeolitite | 27.08 b | 2.06 b | 1.08 b |
| 30/09/2016 | 2—Farm | 29.58 b | 2.32 b | 1.17 b |
| | 3—Untreated control | 100.00 a | 100.00 a | 100.00 a |

* Percentage of plants with infected bunches. Statistical analyses were performed according to Tukey's honest significant difference (HSD) test. For each column, assessment and year, different letters represent significant differences at $p$ = 0.05.

3.1.2. The 2016 Trial

In the first assessment carried out on May 30 (BBCH 60), after the frequent May rainfalls, 11.67% of the untreated control vines showed symptoms on the leaves (Table 2, Table 3 and Table S2). In the second assessment carried out on June 30 (BBCH 75), after the June rainfalls, as high and frequent as May, the vines with foliar symptoms were 89.17% (Table 2, Table 3 and Table S2). The percentages of infected leaves increased from 0.33% (first assessment) to 4.88% (second assessment) (Table 3). Similar increases between the first and second assessment were also recorded in the percentages of infected surface and fertile infected surface on infected leaves, which in turn were 5.26% (Table 3).

As in 2015, no infected clusters were observed in the first two assessments carried out in 2016.

In the first two assessments, all applications significantly and to the same extent reduced infections compared to the untreated control (Table 3). In the second assessment, the percentages of vines with symptoms on the leaves were 4.17 and 3.75, in chabasite-rich zeolitite and farm treatments, respectively (Table 3). In both treatments, the percentage of infected leaves, infected surface and fertile infected surface was always lower than 0.1% (Table 3).

In the third assessment (10 August, BBCH 83), after the rains of July and August 1st, all untreated control plants showed symptoms of berry brown rot, with the percentages of infected bunches and infected berries being 30.36% and 9.46%, respectively (Table 2, Table 4 and Table S2). After the rains of August and September, all plants and bunches of the untreated controls became fully infected in the fourth assessment (30 September), at BBCH 89 (Table 2, Table 4 and Table S2).

The 2016 season was characterized by a strong pressure of the disease. Therefore, in the third assessment, plots treated with chabasite-rich zeolitite and farm products showed about 20% of vines with infected bunches (Table 4). However, the infected bunches and infected berries did not exceed 1.56% and 0.32%, respectively (Table 4). The percentages of all parameters detected in the fourth assessment on the treated vines were still low, particularly when compared with 100% of the damage detected in the untreated control at harvest (Table 4, Figure 1).

*3.2. Effect of the Applications against Downy Mildew on Yield Quantity, Grape and Wine Composition*

In both trial years, no differences among treatments were found both in the quantity of grape yield per plant and for the other investigated parameters in grapes (Table 5). Only the 2015 grapes of the untreated controls showed a non-statistical decrease in both soluble solids (about 0.5° Brix) and total polyphenols of about 60–70 mg $L^{-1}$ compared to the other treatments (Table 5).

The decrease in soluble solids observed in the grapes of the untreated control in 2015 was still evident in the wines obtained from these grapes as alcohol content, although the difference was still not significant and even less clear (Table 6). The slightly lower polyphenol content in the grapes of the untreated control compared to the grapes of the farm treatment was no longer observed in the wines of the two treatments (Table 6).

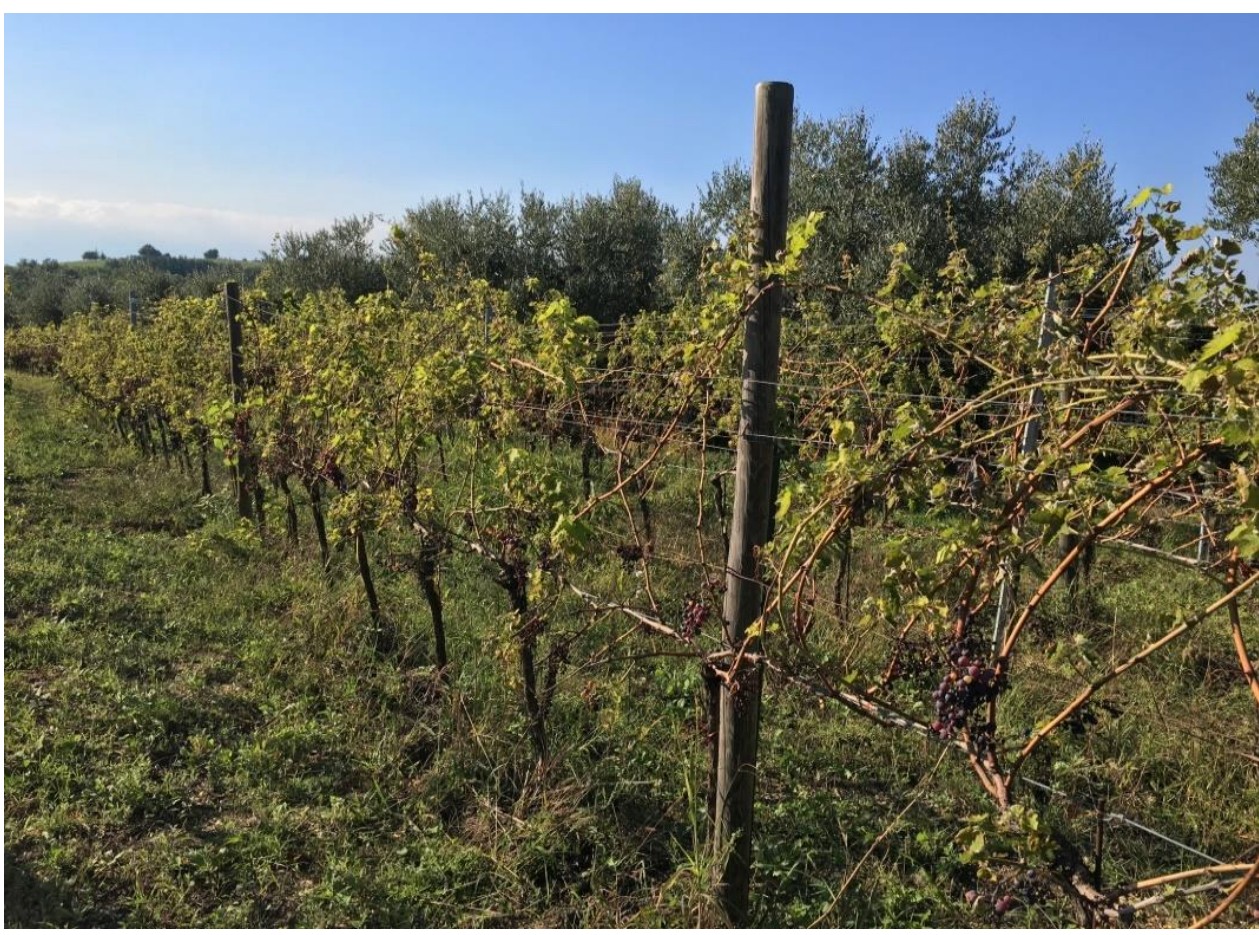

**Figure 1.** Untreated control plants at the end of the 2016 season. Bunches were completely damaged by downy mildew infection.

In both trial years, after 3 months of ageing, wines did not show any difference among treatments regarding the alcohol content, pH and total acidity (Table 6). Conversely, in 2015, statistically higher polyphenol content (3428 mg $L^{-1}$) was observed in the wine obtained from the plots treated with chabasite-rich zeolitite compared to the plots treated with farm products and to the untreated control plots, 3322 and 3338 mg $L^{-1}$, respectively (Table 6). Conversely, in 2016, the difference in the polyphenol content between chabasite-rich zeolitite and farm treatments (3122 and 3068 mg $L^{-1}$) was not statistically significant.

The wines obtained from the chabasite-rich zeolitite plots varied from those of the other plots, achieving higher color intensity (IC3), in the order of 2 points in 2015 and 1 point in 2016 (Table 6). Again, these results differed statistically in 2015, but not in 2016.

**Table 5.** Mean yield and main grape composition parameters recorded in 2015 and 2016 at harvest in Montepulciano vines of the Ari vineyard treated with Italian copper chabasite-rich zeolitite and farm products.

| Treatment | Yield (Kg vine$^{-1}$) | | Soluble Solids ($^\circ$ Brix) | | pH | | Total Acidity (g L$^{-1}$) | | Total Polyphenols (mg L$^{-1}$) | |
|---|---|---|---|---|---|---|---|---|---|---|
| | 2015 | 2016 | 2015 | 2016 | 2015 | 2016 | 2015 | 2016 | 2015 | 2016 |
| Chabasite-rich zeolitite | 5.55 a | 5.04 a | 24.18 a | 23.77 a | 3.43 a | 3.34 a | 6.08 a | 6.57 a | 1385 a | 1115 a |
| Farm | 5.59 a | 5.08 a | 24.19 a | 23.73 a | 3.41 a | 3.31 a | 6.11 a | 6.59 a | 1373 a | 1122 a |
| Untreated control | 5.25 a | n.r. | 23.64 a | n.r. | 3.33 a | n.r. | 6.16 a | n.r. | 1309 a | n.r. |

n.r.: not recorded for the absence of a healthy yield. Statistical analyses were performed according to Tukey's honest significant difference (HSD) test. For each column, different letters represent significant differences at *p* = 0.05.

**Table 6.** Mean composition parameters for Montepulciano wines of the Ari vineyard treated with Italian copper chabasite-rich zeolitite and farm products, recorded in 2015 and 2016, after 3 months of ageing.

| Treatment | Ethyl Alcohol (% vol.) | | pH | | Total Acidity (g L$^{-1}$) | | Total Polyphenols (mg L$^{-1}$) | | IC$_3$ | |
|---|---|---|---|---|---|---|---|---|---|---|
| | 2015 | 2016 | 2015 | 2016 | 2015 | 2016 | 2015 | 2016 | 2015 | 2016 |
| Chabasite-rich zeolitite | 14.33 a | 14.02 a | 3.56 a | 3.44 a | 7.28 a | 7.72 a | 3428 a | 3122 a | 15.3 a | 12.7 a |
| Farm | 14.29 a | 14.00 a | 3.54 a | 3.44 a | 7.36 a | 7.77 a | 3322 b | 3068 a | 13.3 b | 11.6 a |
| Untreated control | 14.01 a | n.r. | 3.50 a | n.r. | 7.28 a | n.r. | 3338 b | n.r. | 13.5 b | n.r. |

n.r.: not recorded for the absence of a healthy yield. Statistical analyses were performed according to Tukey's honest significant difference (HSD) test. For each column, different letters represent significant differences at *p* = 0.05. IC3: wine color intensity.

## 4. Discussion

Increasing environmental concerns and the occurrence of resistance to fungicides are probably the main causes of the decreasing use of synthetic products [26]. Therefore, studies on beneficial microorganisms and natural substances are being carried out. Specific studies are focused on the reduction in the use of cupric fungicides to limit the environmental impact of copper in the soil, following the EC regulations 1981/2018 [24,26–29]. In particular, some studies assessed alternatives to copper or new formulations with lower copper content [30].

In this regard, the activity of natural formulations based on mineral powders such as chabasite-rich zeolitites was evaluated. Zeolitites have physicochemical characteristics that hinder the establishment of infections of pathogens favored by high humidity and wetting [31,32]. Two different Italian zeolitites, chabasite-rich zeolitite and copper chabasite-rich zeolitite, have been successfully evaluated before in the simultaneous control of grapevine grey mold, sour rot and *Lobesia botrana* [16,17].

Copper chabasite-rich zeolitite differs from chabasite-rich zeolitite due to the presence of copper, such as sulphate pentahydrate and hydroxide, incorporated inside the tetrahedral zeolitite structure and is therefore not easily washable. Both zeolitites controlled infections of grey mold and sour rot, although copper chabasite-rich zeolitite showed higher activity than copper-free zeolitite. This could have been due to the activity of copper against bunch rots, but also due to the size of the particles, which are finer than copper-free zeolitite, improving the uniformity of the deposit [16,17].

These positive results encouraged the present study carried out in 2015 and 2016, on the activity of copper chabasite-rich zeolitite in comparison with a farm strategy against grapevine downy mildew.

The numerous infective rains recorded in 2015 and 2016 made these seasons ideal for verifying the efficacy of the different control strategies.

The preventive control strategy, based on applications carried out before a putative infective rainfall, proved to be a reliable approach.

In the case of the copper zeolitite, the preventive strategy was possible thanks to the persistence and adhesiveness of the product on the plant [16,17].

Similarly, in the farm treatment, all applications were carried out before the expected infective rains, preferring preventive rather than curative applications, to avoid the resistance of fungal populations to systemic fungicides [33].

In both years, the applications of either copper zeolitite or systemic fungicides used in farm strategy were equally effective in the growth stages with high downy mildew infection risk (53 BBCH–73 BBCH). In the following growth stages, the high efficacy provided by the applications of a tank combination of Bordeaux mixture with copper sulphate complexed with amino acids was comparable to that obtained with copper zeolitite. This result is particularly important for the significant reduction of copper metal: for each application, the Bordeaux mixture + copper sulphate complexed with amino acids released 562 g ha$^{-1}$ of copper metal versus 240 g ha$^{-1}$ for copper zeolitite.

The excellent activity of the Bordeaux mixture was probably increased by the copper sulphate complexed with amino acids, able to penetrate thanks to a passive membrane crossing mechanism, typical of organic substances, known as the "smuggling" effect, giving the product a cytotropic effect [34].

The activity of copper zeolitite against downy mildew is probably due to its water adsorption capacity and to the consequent reduction in humidity in the canopy [31,32,35,36]. The application of zeolitite causes the formation of a microscopic and hydrophilic layer of mineral particles, capable of both absorbing condensed water and eliminating free water [36,37]. This layer acts as a physical barrier, probably reducing the germination of *P. viticola* spores [38]. Therefore, the properties of copper zeolitite were effective not only in reducing the infections of grey mold and sour rot [16,17], but also towards *P. viticola*.

In addition, the copper was probably less washed out because of the incorporation in the zeolitite structure. Furthermore, the activity of the copper zeolitite could be enhanced

by its adhesiveness on the treated surfaces, compared to the losses occurring with the Bordeaux mixture application. Therefore, copper zeolitite appears to be effective and less prone to drift losses in the environment.

In both trial years, the applications of copper zeolitite did not decrease the quantity and quality of grape and wine compared to what was detected with the farm strategy. These results are in agreement with what was found in previous studies on the yield of vines treated with copper zeolitite or pure zeolitite once applied towards grey mold and sour rot [16,17].

In 2016, the levels of the analyzed quality parameters of grape and wine were lower than those recorded in 2015, with the exception of the higher total acidity. These results are likely to be attributed to the 2016 season, characterized by a greater number of rainy days and lower temperatures during ripening, compared to 2015. These low temperatures during ripening may have led to reductions both in photosynthetic capacity and consequently in sugars and other parameter levels, while the total acidity increased.

In 2015, the wine obtained from vines treated with copper chabasite-rich zeolitite showed contents of total polyphenols and levels of color intensity statistically higher than the wine obtained from both the farm strategy and the untreated control vines. A similar result had already been observed in a previous study, carried out in the 2017, in the wine obtained from cv. Montepulciano vines treated with pure zeolitite [16]. In this previous study, carried out in a particularly dry and hot season, the activity of zeolitite on quality parameters of wine was clear, increasing the polyphenol contents to levels similar to those usually recorded in Montepulciano cultivar [39] and higher compared to those recorded in the other treatments [16]. In the present study, in the 2015 season, which was rainier and colder than the 2017 season, the bunches treated with copper zeolitite showed an increase in total polyphenols and color intensity, compared to the other treatments, but less noticeable compared to what was recorded in the 2017 study. A protective effect of inert dust similar to that of kaolin was hypothesized. Kaolin is capable of reflecting solar radiation, decreasing the temperature of the treated plant surface, hence favoring secondary metabolism and the biosynthesis of phenolic compounds [37,40,41]. The exposure of bunches of black berry cultivars to high temperatures reduces the synthesis of phenolic compounds and color intensity in the wine [42–44].

As reported above, the 2016 temperatures occurring during the ripening phase were lower than the 2015 temperatures, with a higher frequency of rainfall too; therefore, the effect of zeolitite on the quality parameters was no longer detectable, since the levels of color intensity and the content of total polyphenols did not statistically differ between zeolitite and farm strategy treatments. It may not be excluded that the decrease in both sugar levels in bunches and ethyl alcohol and total polyphenols in wines, assessed in 2016 compared to 2015 and probably linked to a lower photosynthetic activity during the maturation process, is the cause of the absence of significant differences among treatments. On the whole, the protective effect of the bunch by chabasite-rich zeolitite towards high temperatures was further confirmed.

## 5. Conclusions

To our knowledge, this is the first study that demonstrates the activity of copper chabasite-rich zeolitite against downy mildew of grapevine. This result is of particular interest since the activity of the product was already proved in previous studies towards grey mold and sour rot. Therefore, in some growth stages, just one application could have the potential to simultaneously control sour rot, grey mold and downy mildew, thus reducing the quantities of copper used as well. Furthermore, this possible reduction is due not only to the reduction in the number of applications but also to the low percentage of copper in the zeolitite compared to traditional copper-based products.

Copper chabasite-rich zeolitite should therefore be considered a low environmental impact product.

The activity of copper chabasite-rich zeolitite was high at all growth stages, including those at high infection risk, and is comparable to the activity of synthetic cytotropic or systemic fungicides, which in turn can be used less than usual, thus reducing the risk of resistance to pathogens.

The reduction in the copper amount is particularly important in the case of organic viticulture, where the use of synthetic products is forbidden. A similar efficacy between copper chabasite-rich zeolitite and standard/conventional strategies against downy mildew is also important because studies hypothesized a decrease in production during the transition from conventional to organic viticulture [45]. In this cited study, the production decrease was associated with the ban on using synthetic/systemic fungicides which are very effective against downy mildew. On the contrary, the present study demonstrated that copper chabasite-rich zeolitite can preserve the production quality and quantity as much as synthetic/systemic fungicides.

Furthermore, the protection of the bunch from high temperatures can be considered an added value in the use of copper chabasite-rich zeolitite, particularly in hot and dry seasons, which are more frequent as a consequence of climate change. Lowering the temperature of the berries could preserve the biosynthesis of phenolic compounds and facilitate the achievement of adequate levels of coloring intensity in wines. However, it is very important to follow the fifteen-day interval between the last application and the harvest, in order to avoid a decrease in wine polyphenol content and color intensity, as verified in a previous study [16].

**Supplementary Materials:** The following supporting information can be downloaded at: https://www.mdpi.com/article/10.3390/agronomy12071528/s1. Table S1: Rainfall, temperature and leaf wetness periods on rainy days at the Ari vineyard in 2015; Table S2: Rainfall, temperature and leaf wetness periods on rainy days at the Ari vineyard in 2016.

**Author Contributions:** Conceptualization, F.C.; methodology, F.C. and L.S.; software, G.P.; validation, F.C., S.D.M. and G.P.; formal analysis, G.P.; investigation, F.C. and L.S.; resources, F.C.; data curation, F.C., E.G.M. and G.P.; writing—original draft preparation, F.C.; writing—review and editing, F.C. and S.D.M.; visualization, F.C. and S.D.M.; supervision, F.C. and S.D.M. All authors have read and agreed to the published version of the manuscript.

**Funding:** This research received no external funding.

**Institutional Review Board Statement:** Not applicable.

**Informed Consent Statement:** Not applicable.

**Data Availability Statement:** Not applicable.

**Conflicts of Interest:** The authors declare no conflict of interest.

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
