# Peer review of "Control of Grapevine Downy Mildew by an Italian Copper Chabasite-Rich Zeolitite"

_agronomy, doi:10.3390/agronomy12071528_

Round 1

Reviewer 1 Report

This article is interesting, useful, and well prepared.

Introduction is in line with the in Instructions for the Authors. The methodology is also corresponding to the experimental part points of interest. Results are representative and comprehensive with the remark already done in the General comment.

Discussion is also appropriate, but a more precise direction of future research should add more value.

References are in line with the scientific demonstration of the main issues.

This is a carefully done study and the findings are of considerable interest. This article is structured, and the results and discussed, however I suggest the following revisions before acceptance to highlight its novelty.

- Comment and suggested  improvement:   

1) The abstract needs to be restructured as more emphasis needs to be placed on novelty of work an technical achievements. I suggest move some first sentences of Abstract to Introduction and short the sentences in Abstract. Abstract better to be included by more findings of the work to have a more quantitative Abstract.

2) The Introduction requires highlighting the originality of this work.

3)English writing must be improved.

It is an interesting research work and it is recommended to publish it as a first from a series with incoming findings.

Author Response

First of all, we would like to thank you for taking the time to review our manuscript.

1) The abstract has been restructured by adding some findings of the work.

2) At the end of Introduction, at line 115-116, a sentence on the originality of this work was introduced.

3) English writing has been improved.

4) The Discussion was completely rewritten also taking into account your suggestion.

Reviewer 2 Report

The manuscript by Calzarano et al. shows results of a two-year field trial comparing the efficacy of a rock flour - copper product to a  conventional farm crop protection strategy and a non-treated control against Plasmopara viticola on grapevine. Since the use of chemical synthetic fungicides (conventional/integrated production) and copper fungicides (conventional/integrated production AND organic production) have to be reduced, such a product could be interesting for practice. The practice trials performed by the authors are sound. Yet, for a better interpretation of the results, in my opinion, two controls are missing: (i) a traditional copper product at the same copper rates and (ii) a control with rock flour without copper. (i) would allow to assess the additional value of the new product compared to already existing copper fungicides which are widely used. Especially in organic agriculture, crop protection strategies in many countries are widely based on these products and have been shown to allow an effective protection. (ii) would allow to assess whether further copper reduction is possible with the new product, which would be highly beneficial.

Besides these general remarks, the quality of the manuscript in my opinion needs improvement. Especially introduction and discussion (and, in consequence, also the summary) have to be significantly improved, while presentation of the results is more or less ok. Please see detailed comments in the attached manuscript.

Author Response

First of all, we would like to thank you for taking the time to review our manuscript.

Your suggestions have been much appreciated for the improving of the manuscript.

Please note that the corrections refer to the line numbers of the manuscript version with your suggestions.

Abstract: Integrations and corrections were carried out.

Introduction

Line 35: The sentence was rephrased: “The disease causes the most serious yield losses affecting inflorescences, bunches and berries”.

Line 40: “Until the end of flowering” instead of “In these phenological stages” was written.

Line 48: The sentence was rephrased: “Contact fungicides are mainly based on copper and dithiocarbamates and can protect the foliar surface by preventing the zoospore germination and leaf penetration [7,8].”

Line 59: a reference was added

Line 53: “carry out” instead of “improve” was written.

Lines 64-66: The sentence was rephrased: “The development of products based on microorganisms in grapevine disease control increased, but in many cases, these products have provided inconsistent…”

Lines 69-71: According to your suggestion, the sentence was deleted.

Lines 74-76: At lines 75-76 it was reported that a product aimed at simultaneously controlling different pathogens is “simpler” to introduce in a disease control strategy. This is due to the possibility of reducing the number of treatments with significant economic and environmental advantages. The sentence at lines 75-76 does not refer to the possibility of controlling insects; in the following sentence, at lines 78-80, the use of pure zeolite is mentioned only as an example of simultaneous control of different grapevine diseases, also pointing out the activity of pure zeolitite towards Lobesia botrana.

Line 80: the activity was also directed towards grapevine moth.

Line 82: Yes, it is the correct terminology.

Lines 84-86: explanations on the use of zeolites in agriculture was added: "Thanks to these properties, zeolitite improved soil physico-chemical characteristics as: i) increase in CSC and water retention, maintenance of permeability in sandy soils, and increase of aeration and permeability in clayey soils; ii) temporary capture by cationic exchange of K and N-ammonia supplied by a synthetic or natural fertilizer, and their slow and gradual release; iii) greater availability of nutrients for crops and lower losses due to leaching (reference number in the corrected version of the manuscript: 20)."

Line 90: replaced “highest” with “high”.

Line 96: described in the materials and methods section.

Line 100: The sentence was rephrased: “For each of the two years of investigation, the study compared a standard integrated/conventional strategy, normally provided in the vineyard, with a strategy based only on copper chabasite-rich zeolitite applications”.

Materials and methods

Line 112: replaced “by” with “of”

Line 118: We have written: “Following the weather forecast, chabasite-rich zeolitite applications were carried out before each rainfall…”. We have not followed a forecast model. The applications were carried out before expected rains predicted by the weather forecast. All the applications were carried out on the same day as shown in the treatment plan.

Line 117: replaced “endotherapic products” with “systemic fungicides”.

Line 121: “by means of hydraulic or pneumatic drilling machines” was added.

Line 136, Treatment plan: First of all, thanks for the suggestion about assessing the effectiveness of both pure zeolitite and conventional copper fungicide at the copper rates used in the copper chabasite-rich zeolitite, thereby reducing the amount of copper used in the vineyard. It can be considered in further specific studies on the basis of the achievements obtained in the present work.

The applications of pure chabasite-rich zeolitite were compared with those of copper chabasite-rich zeolitite in a previous work focused on the control of grey mould and sour rot (See in the “References”, Calzarano et al., 2020, number 17 in the corrected version).

In the present work the standard/conventional strategy was only compared with the copper chabasite-rich zeolitite, primarily because of the well-known activity of copper against downy mildew, and also for the higher activity showed by copper chabasite-rich zeolitite (compared zeolitite without copper) against grey mould, being the copper effective also towards this pathogen.

The possibility of using copper chabasite-rich zeolitite in applications focused on downy mildew (where copper compounds play a primary role) to replace cupric products characterized by a higher copper content, or even a standard/conventional strategy based on systemic fungicides, represents a first significant improvement in the reduction of copper for a sustainable control strategy.

Tables 1 and 2: replaced “toward” with “targeted again”.  

Tables 1 and 2: I agree with your comment (our mistake). The percentage concentration of the active ingredients of the products was reported by mistake, instead of the rate (in the case of Fenamidone + Fosetyl Al + Iprovalicarb). The tables were corrected by inserting the correct rate and reporting the percentage concentration of the products as foot note.

Table 1: The last application was carried out on September 19. The harvest dates were reported in the text. Moreover, a minimal time of 15 days must elapse between the last zeolitite application and the harvest, was also reported in the text.

Line 150: replaced “surveys” with “assessments”.  

Line 153: each of” was deleted

Line 156: The number of bunches normally produced by the vineyard plants is 16

Line 178: References of berry and wine analyses were reported at lines 189-191.

Results

Line 205: after “fertile infected surface”, “(percentage leaf area with sporulating lesions)” was added.

Line 234: Yes, only slightly

Line 236: In our opinion, all assessments should be considered important, so we think that the presentation of the results could remain as it is.

Table 3: replacedefficacywith “incidence and severity.

Table 3: “infected leaf surface”; added as foot note: “percentage leaf area with sporulating lesions”.

Table 4: replacedefficacywith “incidence and severity.

Figure 1: A picture was taken on vines treated with copper chabasite-rich zeolitite and showed the difference between these plants and those of untreated control, but we didn’t use it because unfortunately the bunches are not in the foreground because they are covered by the foliage.

Table 5: In 2015, about 15 % of untreated control vines did not show any symptoms (vs. 86.25 % of infected plants, Table 4). The bunches were collected only from the 15 % of untreated control vines without downy mildew infection. The bunches were not collected from the diseased vines of untreated control (86.25 %) because of severely attacked. For this reason, the collected bunches of the untreated control plants had no decrease in production compared to the ones of the other treatments.

Discussion

Lines 318 -324: The period at lines 318-324 has been rewritten.

Line 330: “before” was added

Line 337: “improving the uniformity of the deposit” was added after copper free zeolitite”.

Comment at line 344: The product can be used in organic viticulture and also in integrated or conventional viticulture.

Lines 358-360: It’s true, the various cupric products show excellent activity; however, as stated in the cited study (reference 34), the addition of copper sulphate complexed with amino acids improves the performance of the Bordeaux mixture. Therefore, in our opinion, the sentence could be maintained as it is.

Comment at line 361: The comparison between the release of copper by zeolitite and by the other cupric product is reported in Material and Methods at lines 121-128, and in the Discussion at lines 355-357.

Line 370: “probably” after “was” was added

Line 371: “is” was replaced by “could be”

Lines 386-389: The entire period was deleted.

Lines 390-415: The entire period was rewritten.

Conclusions

Line 421: To our knowledge, this is the first study against downy mildew of grapevine. The study carried out in 2017 assessed the activity of zeolitites against grey mould and sour rot.

Line 422: “could have the potential to simultaneously control…” was added

Line 424: We added: “possible”: “Furthermore, this possible reduction…” was added

Line 425: The use of copper at lower rate can be the target of one more specific study.

Line 427: The amount of copper released by the copper zeolitite is considerably lower than that released by Bordeaux mixture, as reported in materials and methods section.

Line 429: “both” was deleted.

Lines 433: The sentence was changed: “The reduction of copper amount is important in the case of organic viticulture…”

Lines 433-438: The period was rephrased: “A similar efficacy between copper chabasite-rich zeolitite and standard/conventional strategies against downy mildew is important also because studies hypothesized a de-creasing of production in the transition from conventional to organic viticulture [46]. In this study the production decreasing was associated with the ban on using synthet-ic/systemic fungicides very effective against downy mildew. On the contrary, the present study demonstrated that copper chabasite-rich zeolitite can preserve the production qual-ity and quantity as much as synthetic/systemic fungicides.

Reviewer 3 Report

The paper can be published with minor revisions and my recommendations are:

- some English errors must be corrected

- improve the abstract with some obtained results

- for the statement "The copper metal content of the product is 6%." please provide some scientific evidences (analysis, etc) if it is possible. The ref 17 is not concludent

Author Response

First of all, we would like to thank you for taking the time to review our manuscript.

1) The English errors has been corrected.

2) The abstract has been restructured by adding some findings of the work.

3) Copper chabasite-rich zeolitite is a commercial formulation (Menorame ®, Agrisana s.r.l., Latina, Italy). No specific analyses on the percentage of copper content are available.

4) The reference 17 is reported because of the mineralogical composition of the product was described.
